# Interplay of Angiotensin Peptides, Vasopressin, and Insulin in the Heart: Experimental and Clinical Evidence of Altered Interactions in Obesity and Diabetes Mellitus

**DOI:** 10.3390/ijms25021310

**Published:** 2024-01-21

**Authors:** Ewa Szczepanska-Sadowska

**Affiliations:** Department of Experimental and Clinical Physiology, Laboratory of Centre for Preclinical Research, Medical University of Warsaw, 02-097 Warsaw, Poland; eszczepanska@wum.edu.pl

**Keywords:** angiotensin, diabetes mellitus, heart failure, hypertension, hypoxia, insulin, obesity, vasopressin

## Abstract

The present review draws attention to the specific role of angiotensin peptides [angiotensin II (Ang II), angiotensin-(1-7) (Ang-(1-7)], vasopressin (AVP), and insulin in the regulation of the coronary blood flow and cardiac contractions. The interactions of angiotensin peptides, AVP, and insulin in the heart and in the brain are also discussed. The intracardiac production and the supply of angiotensin peptides and AVP from the systemic circulation enable their easy access to the coronary vessels and the cardiomyocytes. Coronary vessels and cardiomyocytes are furnished with AT1 receptors, AT2 receptors, Ang (1-7) receptors, vasopressin V1 receptors, and insulin receptor substrates. The presence of some of these molecules in the same cells creates good conditions for their interaction at the signaling level. The broad spectrum of actions allows for the engagement of angiotensin peptides, AVP, and insulin in the regulation of the most vital cardiac processes, including (1) cardiac tissue oxygenation, energy production, and metabolism; (2) the generation of the other cardiovascular compounds, such as nitric oxide, bradykinin (Bk), and endothelin; and (3) the regulation of cardiac work by the autonomic nervous system and the cardiovascular neurons of the brain. Multiple experimental studies and clinical observations show that the interactions of Ang II, Ang(1-7), AVP, and insulin in the heart and in the brain are markedly altered during heart failure, hypertension, obesity, and diabetes mellitus, especially when these diseases coexist. A survey of the literature presented in the review provides evidence for the belief that very individualized treatment, including interactions of angiotensins and vasopressin with insulin, should be applied in patients suffering from both the cardiovascular and metabolic diseases.

## 1. Introduction

In each vascular bed, the blood flow depends on perfusion pressure, the rheologic properties of the blood, the structural and functional properties of the vascular wall, and the action of vasoconstrictory and vasodilatory factors produced locally or inflowing from the systemic circulation. The fundamental role of the heart in the maintenance of blood circulation requires the particularly precise regulation of the coronary blood flow (CBF). Rhythmic contractions of the heart exert direct mechanical effects on the coronary vessels that are associated with the production of metabolic factors in the cardiac myocytes and the smooth muscle cells of coronary vessels. The local cardiac vasoactive compounds regulate CBF in concert with cardiovascular compounds inflowing from the systemic circulation, and with the neurogenic control of the heart by the autonomic nervous system [1]. Considerable evidence shows that the mechanical performance of the cardiac muscle and the production of the vasoactive factors are significantly affected in cardiovascular and metabolic diseases [2,3,4]. Among factors that regulate CBF and cardiac metabolism, are components of the renin–angiotensin system (RAS), vasopressin system (VS), and insulin [5,6,7,8,9]. In addition, the activation of RAS generates oxidative stress, which causes the overproduction of reactive oxygen species (ROS), as well as significant disturbances of sarcolemma, sarcoplasmatic reticulum, and myofibrils that contribute to the development of cardiomyopathy [10,11,12,13]. An increasing number of studies provide evidence that RAS and VS participate in the regulation of metabolism and closely cooperate with insulin in this respect. Insulin resistance, cerebral glucose hypometabolism, cardiac autonomic neuropathy, and cardiovascular disturbances belong to serious complications of the diabetes mellitus of type 1 (T1DM) and type 2 (T2DM), alongside prediabetes (pre-DM) and metabolic (MetS) syndromes [14]; however, the causative links of these disorders are not yet fully understood [15,16,17,18].

The present review is a survey of experimental and clinical studies, providing evidence that angiotensin peptides and vasopressin (AVP) play important roles in the regulation of coronary circulation and cardiac muscle properties. The cooperation between these peptides and insulin is emphasized, drawing attention to the aberrant interactions of RAS and AVP with insulin in diabetes mellitus and other metabolic disorders.

## 2. Renin-Angiotensin System

### 2.1. Components of Renin-Angiotensin System

The RAS includes renin, angiotensinogen, and angiotensin peptides. Renin is a highly potent enzyme, acting on the N-terminal of angiotensinogen (AGT), which, in humans, is a molecule that is composed of 485 aminoacids. Renin removes a decapeptide angiotensin I [(Ang I, Ang-(1-10)] which can be converted to either the highly active angiotensin II [(Ang II, Ang-(1-8)] via the angiotensin-converting enzyme 1 (ACE1), or to the less active Ang-(1-9) via the angiotensin-converting enzyme 2 (ACE2). Ang-(1-7) can be cleaved either from Ang-(1-9) by ACE1 or from Ang-(1-8) by ACE2 (Figure 1). Other components of the RAS system [Ang III, Ang-(2-8), Ang IV, Ang-(3-8), Ang-(1-5), Ang-(5-8), Ang-(1-12)] are produced by enzymatic transformations that engage carboxypeptidases and endopeptidases.

Angiotensin peptides interact with AT1R (in some species, AT1aR and AT1bR subtypes), AT2R, and MasR receptors, which are coupled with G proteins. The activation of AT1R is mediated by multiple enzymatic pathways, including phospholipase C (PLC), phospholipase D, phospholipase A2 (PLA2), nicotinamide adenine dinucleotide phosphate (NADPH) oxidase, and metalloproteinases. The activation of these tracks results in the stimulation of highly active proteins, such as cyclooxygenases, lipooxygenases, cytochrome P450 enzymes, mitogen-activated protein kinase (MAPK), c-Jun N-terminal kinase (JNK), extracellular-signal-regulated protein kinases 1 and 2 (ERK1/2), and transcription factors (NF-κB, AP-1, HIF-1α). The stimulation of AT2R results in the activation of phosphotyrosine phosphatases and the inactivation of protein kinases, potassium channels, PLA2, and arachidonic acid derivatives. Ang-(1-7) activates the Mas receptor (MasR) and the ACE2–Ang-(1-7)–MasR axis [19,20].

### 2.2. Cardiac Effects of Ang II

Thus far, the regulation of the functions of cardiomyocytes and coronary vessels by Ang II have been assessed, mainly in experimental studies.

#### 2.2.1. Action of Ang II on Cardiac Muscle

Angiotensin receptors are present in both the coronary vessels and the cardiac muscle (Figure 2). The studies demonstrating the autocrine release of Ang II from the mechanically stretched cardiomyocytes suggested that Ang II may be produced in the heart and may contribute to the development of stretch-induced hypertrophy [21]. The essential role of AT1R in the appropriate functions of the heart was confirmed in studies conducted on control mice and mice with AT1aR and AT1bR knockout; the knockout resulted in atrophic changes of the myocardium, in the abolishment of the Ang II–induced decrease of coronary blood flow, and in the reduction of the left ventricle pressure [22].

It is likely that Ang II induces the apoptosis of cardiac cells through the actions exerted in mitochondria. The exposure of neonatal cardiomyocytes to a high concentration of Ang II elicited mitochondrial damage with the downregulation of the NADPH dehydrogenase subunit, which is a component of the electron transport chain [23,24]. Furthermore, it has been shown that Ang II enters the mitochondria, where it stimulates NADPH oxidase 4 (NOX4) and promotes electron leakage and ROS production, which may cause damage of the mitochondrial DNA. In the mitochondria, Ang II also stimulates other destructive processes, such as the oxidation of components of the membrane permeability transition pore, and the activation of the mitochondrial ATP-sensitive K^+^ channel. Accordingly, it has been suggested that the above processes play a pivotal role in Ang II-induced cardiac hypertrophy and endothelial dysfunction, and that the inappropriate mitochondrial action of Ang II contributes to the development of cardiac and metabolic diseases [24,25,26].

#### 2.2.2. Action of Ang II on Coronary Vessels

Historically, the first extensive study exploring the effects of the systemic administration of Ang II on coronary vessels was performed in 1976 by Giacomelli et al. [27] on Holtzman rats. The study demonstrated that Ang II, delivered in a hypertensive dose (1.7 μg/min/kg) for 4 h, induced the injury of coronary vessels; this was manifested by the increased permeability of the epicardial arteries and lesions of intramural arteries and arterioles. Next, experiments on Wistar Kyoto rats showed that the infusion of Ang II at the same rate for 2 h increased blood pressure and elicited the vasoconstriction of intracardiac intramural arteries and arterioles which were associated with endothelial cell vacuolization, smooth muscle cell fragmentation, and necrosis [28]. Experiments on trained rats revealed that the administration of Ang II prominently reduced the density of the left ventricle (LV) vessels of the trained animals performing repetitive exercises for 10 weeks [29].

Human in vitro studies. Experiments on the cultured coronary artery smooth muscle cells of humans showed that Ang II enhances the migration and proliferation of these cells and that this process can be blocked by AT1R antagonist valsartan [30]. There is evidence that the growth-promoting action of Ang II on the smooth muscle cells of human coronary vessels requires activation of the mammalian target of the rapamacine (mTOR)-sensitive signaling pathway, as well as of phosphatidylinositol 3-kinase, p70(s6k), and eukaryotic inhibitor factor-4E [31]. Human coronary artery endothelial cells (HCAECs) also respond to moderate concentrations of Ang II with the upregulation of the proteins that promote tube formation, such as signal transducers and the activator of transcription 3 (STAT3) and mir-21; it can also be presumed that Ag II is involved in the process of vasculogenesis [19,32].

Experimental evidence for the role of intracardiac RAS. With regard to the potential role of intracardiac Ang II in coronary vessels, there is evidence for the presence of some of the components of RAS in the heart. Experiments on cultured bovine aortic endothelial cells provided evidence that these cells are able to synthesize and secrete angiotensin peptides, and the authors have suggested that the coronary endothelial cells may possess similar properties [33]. Furthermore, the employment of quantitative in vitro autoradiography in the rat allowed for the demonstration of the presence of ACE in the endothelium and the smooth muscles of the coronary vessels, aorta, and pulmonary artery [34,35,36]. Studies on rat coronary endothelial cells (CES) and vascular smooth cells (VSMC) visualized AT1R in endothelial cells and AT2R in both types of cells. Ang II induced the AT1R-dependent proliferation of VSMC, whereas, in CES, the proliferative effect could be observed after blockade of AT2R. In addition, in CES the antiproliferative action could be observed after the administration of a selective agonist of AT2R (CGP 42112). Altogether, the results suggested that Ang II is able to promote the proliferation of VSMC and can also suppress the proliferation of CES [37]. Both AT1R and AT2R were demonstrated in the smooth muscle cells of porcine coronary artery explants, and both types of these receptors participated in the stimulation of the migration of smooth muscle cells by Ang II [38] (Figure 1).

Connections of Ang II with NO and hypoxia. Strong evidence points to the important role of nitric oxide (NO) in the regulation of coronary vessels by Ang II. Earlier studies indicated that the long-term inhibition of NO synthesis via the administration of N(ω)-nitro-L-arginine methyl ester (L-NAME) in Wistar Kyoto rats enhanced the activation of ACE and induced the increase of the wall-to-lumen ratio and the enhancement of perivascular fibrolysis, thereby exhibiting the remodeling of coronary vessels. These effects were markedly reduced by the administration of an ACE inhibitor (tamocapril) and by the blockage of AT1R [39,40]. Other studies confirmed the important role of Bk and NO in the ACE-mediated vasodilation of coronary vessels [41,42]. In vitro studies on endothelial cells of male Wistar rats showed that Ang II increases NO production in the proliferating cells and that this effect is mediated both by AT1R and AT2R [43]. 

It is likely that the interaction between Ang II and NO plays a role in the adaptation of coronary endothelial cells to hypoxia. Experiments on mice provided evidence that, under hypoxic conditions, Ang II acting on AT2R induces endothelial sprout formation and that this effect is mediated by Bk and NO [44]. Furthermore, it has been shown that the exposure of human coronary endothelial cells to hypoxia increases the phosphorylation of JNK and the activity of hypoxia-inducible factor-1α (HIF-1α). Moreover, together with hypoxia, Ang II increases the secretion of visfatin, which is an adipocytokine with angiogenic properties. The authors found that hypoxia or hypoxia combined with an application of hyperbaric oxygen increases glucose uptake and promotes migration and tube formation in the cells. These effects could be blocked by the AT1R antagonist losartan [45].

Interaction of Ang II and endothelin. There is evidence that Ang II cooperates with endothelin (ET) in the regulation of CBF. The majority of endothelin-1 (ET-1) originates from endothelial cells where its synthesis is stimulated by Ang II, thrombin, and inflammatory cytokines [46,47]. It has been found that cultured endothelial coronary cells co-express Ang II and ET-1 and that the exposure of these cells to isoproterenol or high potassium concentrations increases the co-expression of these peptides, whereas exposure to sodium nitroprusside or S-nitroso-N-acetyl penicillamine (SNAP) decreases Ang II and ET-1 secretion [36]. Other studies have shown that the stimulation of α1 adrenergic receptors in rat cardiomyocytes increases the production of Ang II, which subsequently stimulates the formation of ET-1 through the activation of NADP oxidase [47]. In healthy awake swine, a separate blockade of AT1R and ETA/ETB receptors produced similar vasodilatory responses as did the combined blockade of these receptors; however, afterwards, the myocardial infarction responsiveness to both peptides was significantly altered [48].

#### 2.2.3. Ang II and Epicardial Adipose Tissue

The electrical and mechanical function of the heart as well as coronary blood flow can also be regulated by bioactive compounds generated in the epicardial adipose tissue (EAT) [49]. The pericardial fat is a source of adipokine, anti-inflammatory cytokine (IL-10), pro-inflammatory cytokine (IL-6), and other pro-inflammatory factors, (IFNγ, MCP1) regulating the function of the cardiac muscle and coronary blood vessels. In patients with cardiovascular diseases, the production of ROS in EAT is higher than in the subcutaneous adipose tissue [50]. There is also evidence for activation RAS in the EAT [51,52].

### 2.3. Cardiac Effects of Ang-(1-7) and Other Angiotensins

In many instances, Ang-(1-7), when acting on MasR, exerts similar effects to those observed after the stimulation of AT2R by Ang II, and is involved in the cardioprotective factor (Figure 1). The mechanism of the cardioprotective action of Ang-(1-7) is not yet fully recognized; however, it appears that it depends on the generation of NO [53,54,55,56]. There is also evidence that Ang-(1-7) acts as an ACE1 inhibitor, and that it potentiates the vasodilatory effect of bradykinin via B2 receptors [57]. The possibility that the vasodilatory action of Ang-(1-7) in the coronary vessels is mediated by Bk and NO has been confirmed in experiments on hearts isolated from guinea pigs and Wistar rats [58,59]. Also, other angiotensin peptides [Ang-(1-2), Ang-(1-3), Ang-(1-4), Ang-(1-5)] reduce the pressure perfusion of the rat coronary vessels, and it is likely that their action depends on the stimulation of Mas receptors and the release of NO [60].

### 2.4. Centrally Mediated Effects of Angiotensin Peptides

The local action of Ang II in the heart may be potentiated by the interaction of this peptide with cardiovascular neurons in the central nervous system. Although the systemic Ang II does not cross the blood–brain barrier, it can act in the subfornical organ (SFO), and it can activate neural pathways projecting from the SFO to the paraventricular nucleus (PVN), where it can stimulate AVP secretion and the activation of the sympathetic outflow [61,62,63]. The systemic administration of relatively small doses of Ang II in mice was found to elevate blood pressure, plasma AVP, and endothelin-1 (ET1) levels, and to decrease cerebral blood flow (CBF). As the above effects were significantly reduced by the intra-brain applications of the AT1R antagonist (losartan), the AVP V1aR antagonist (SR49059), and the ROS scavenger (MnTBAP), it is possible that they were mediated by Ang II, which penetrated to the brain, and through the stimulation of the AT1R-elevated release of AVP, the activation of V1aR, and the generation of ROS [64].

### 2.5. Angiotensin Peptides in Regulation of the Heart in Cardiovascular Diseases

Multiple experimental studies provide evidence that the excessive activation of the RAS pathway importantly contributes to the initiation of pathogenic responses in the heart (Figure 2). The most advanced studies refer to the role of Ang II and Ang-(1-7).

#### 2.5.1. Experimental Evidence for the Negative Role of Ang II in the Heart in Myocardial Infarction and Hypertension

Experiments on sham-operated rats and rats with myocardial infarction revealed that the prolonged administration of ACE inhibitor quinapril helps to preserve high energy phosphate metabolism in the infarcted rats, which suggested that endogenously produced Ang II exerts a negative effect on the metabolism of the infarcted heart [65]. The administration of the ACE inhibitor (enalaprilat) in a porcine model of myocardial infarction significantly reduced the area of necrosis and reduced regional cell motion [66]. The positive effects of ACE inhibition and AT1R blockade during myocardial ischemia could also be caused by the increased production of NO and Bk [67,68,69]. Studies on dogs with pacing-induced heart failure revealed that the vasoconstrictory effect of AT1R stimulation on the coronary vessels is markedly attenuated in cardiac heart failure, which suggested the desensitization of the coronary vessels to the vasoconstrictory effect of Ang II. The authors suggested that the desensitization to Ang II may be caused by the accumulation of some vasodilatory compounds, for instance bradykinin and/or NO [70]. Indeed, the inhibition of ACE and blockade of AT1R increased cardiac bradykinin and NO levels, and elevated coronary blood flow in another study on dogs with myocardial ischemia induced by the reduction of coronary perfusion pressure [71]. The AT1R blockade restored endothelial NO synthase, contractile-type myosin heavy chain isoform SM2, and calponin and Gata-6 levels in the coronary vessels of SHR rats [72]. Subsequent studies showed that blockade of AT1R by losartan in rats with the myocardial infarction improved cardiac performance and reversed several negative biochemical processes, such as elevated sarcoplasmic reticulum Ca^2+^ uptake, the activation of Ca^2+^ pump protein, and phospholamban protein, as well as the stimulation of myofibrils (MF) Ca^2+^-activated ATPase, myosin heavy chain (α-MHC) mRNA, and β-MHC mRNA [73]. There is evidence that the myocardial infarction modulates interactions between AT1R and ETA/ETB receptors in the coronary vessels. For instance, it was shown that in healthy awake swine, the separate blockade of AT1R and ETA/ETB receptors produced similar vasodilatory responses as the combined blockade of these receptors [48]; however, 2–3 weeks after the myocardial infarction, the coronary vasodilatory responses to individual blockades of AT1R and ETA/ETB receptors were abolished in spite of that fact that the expression of AT1R and ETA receptors was not altered. In addition, blockade of ETA/ETB receptors in presence of blockade of AT1R was able to produce coronary vasodilation in the infarcted swine. Thus, the authors postulated that under control conditions, Ang II and endothelin act in the coronary vessels as independent vasoconstrictory peptides, whereas the myocardial infarction initiates cross-talk interactions between these compounds at the post-receptor level [48]. The effectiveness of the stimulation of AT1R may depend on the accessibility of the AT1R-associated protein (ATRAP), which is able to enhance the internalization of AT1R from the cell surface to cytoplasm and reduce the action of Ang II [74]. More recently, studies performed on non-culprit arteries harvested from the rabbit model of myocardial ischemia-reperfusion provided evidence that the expression of AT1R was higher in the ischemia-reperfusion group than in the sham group, but expressions of AT1R, connexin 43, and β-tubulin were lower in the ischemic postconditioning group than in the ischemia reperfusion group [75].

There is also evidence for the enhanced involvement of central AT1R in blood pressure regulation in hypertension. Blockade of central AT1R with losartan significantly reduced the hypertension of renin transgenic rats, although it was not effective in normotensive SD rats [76]. Furthermore, it was shown that cardiac sympathetic afferent reflex and renal sympathetic activity were enhanced in SD rats with the myocardial infarction, and that these effects were associated with the elevated stimulation of AT1R receptors by Ang II in the PVN [77].

The central effects of angiotensin peptides may play an especially significant role when the cardiovascular disease is associated with stress. For instance, it has been found that prolonged (2 weeks) restraint stress of C57BL/6J mice enhanced the expression of the mRNA levels of angiotensinogen, TNF-α, IL-6, monocyte chemoattractant protein-1 (MCP-1), insulin receptor substrate (IRS-1), and glucose transporter 4 (GLUT4), and could, in this way, promote the synthesis of angiotensin peptides and cytokines as well as the development of insulin insensitivity [78].

#### 2.5.2. Negative Role of Ang II in the Heart in Human Heart Failure

Positive effects of ACE inhibition and AT1R blockade were also found in human patients with cardiac failure. In patients undergoing coronary arteriography blockade of AT1R with losartan improved epicardial blood flow during a stress-induced cold pressure test and during exercise [79]. Moreover, prolonged therapy with ACE inhibitors promoted the development of collateral circulation in patients with coronary artery stenosis, presumably due to coronary artery angiogenesis [80]. Studies analyzing the expression of ACE in samples of coronary vessels obtained from patients suffering from coronary disease revealed the presence of ACE in the atherosclerotic plaques [81]. In addition, experiments on coronary arterioles obtained from patients suffering from atherosclerosis showed that the incubation of these vessels with an ACE inhibitor (Lisinopril) significantly ameliorated the vasodilatory responses to several endothelium-dependent agonists. The latter effect could be abolished by pretreatment with NO synthase inhibitor [82]. In patients with chronic heart failure, the combined application of an ACE1 inhibitor (enalapril) and losartan for 16 weeks significantly reduced plasma insulin levels, homeostatic model assessment of insulin resistance (HOMA-IR), TNF-α, interleukin-6, and MCP-1 levels. Correlations were found between the reduction of HOMA-IR and decreases of IL-1 and MCP-1, which suggested the significant role of activation ACE1 and AT1R in the regulation of cytokines and insulin secretion [83]. Studies in vitro using the model of endothelium-denuded coronary vessels revealed that levels of AT1R mRNA and AT2R mRNA were lower in the smooth muscle cells of patients with heart failure than in the control subjects [84]. The expression of AT2R was demonstrated in atherosclerotic plaques that were isolated from the internal coronary artery of human patients [85].

#### 2.5.3. Positive Aspects of Action of ACE2 and Ang-(1-7) in Heart Failure

There is evidence that cardiac failure causes the activation of ACE2 and the stimulation of Ang-(1-7) pathways (see Figure 1).

Experimental studies. Experiments on control and infarcted Wistar rats, which were treated or non-treated with selective Ang-(1-7) agonist, Ang-(1-7) antagonist, and with N(G)nitro-l-arginine methyl ester (L-NAME), provided evidence that Ang-(1-7) improves the hemodynamic parameters of the heart and decreases the infarcted area through actions mediated by NO [86]. Prolonged blockade of ACE2 activity in male Wistar rats was found to increase the myocardial infarct size and reduce the left ventricle percentage shortening [87].

Human studies. Experiments on human coronary endothelial cells provided evidence that the stimulation of AT2R increases the expression of ACE2 and that the stimulation of ACE2 and AT2R results in the inhibition of stimulatory effects of TNF α on IkB and NF-kB signaling, which may suggest that the activation of ACE2 and AT2R exerts anti-inflammatory action in these cells [88]. Recently, it has been found that circulating levels of ACE2 and Ang-(1-7) are significantly elevated in patients suffering from coronary artery disease, and that the levels of ACE2 are significantly higher in the female groups than in the male groups of patients [89].

It should be noted that ACE2 acts as a receptor for SARS-CoV-2 (Figure 1) and that the altered activity of ACE2 and/or SARS-CoV-2 may interfere with the development of atherosclerotic lesions in COVID-19 infections; indeed, SARS-CoV-2 viral mRNA is present in atherosclerotic plaques [90]. Moreover, SARS-CoV-2 infection increases the production of pro-atherogenic and proinflammatory cytokines, such as IL-1β and IL-6 [90,91]. Recently, it has been reported that patients infected with SARS-CoV-2 produce autoantibodies that are able to inhibit ACE2 activity and intensify the severity of COVID-19 [92].

#### 2.5.4. Genotypes of ACE and Angiotensin Receptors as Determinants of Coronary Diseases

There is evidence that, to some extent, susceptibility to the myocardial infarction and coronary artery disease may be determined by the specific polymorphism of the ACE genotype. The study comparing the distribution of ACE genotypes in Korean patients suffering from acute coronary syndrome and in healthy subjects indicated that the genotype DD of the ACE gene may be an independent risk factor of acute coronary syndrome [93]. There is evidence for the association of the insertion/deletion (ID) polymorphism of ACE with responsiveness of coronary vessels to NO-mediated vasodilation, i.e., patients with the ACE DD genotype presented significantly lower vasodilatory responses to sodium nitroprusside (SNP) than the other groups of patients [94]. The detection of ACE polymorphism in the leukocytes of patients suffering from coronary artery disease provided evidence that ACE genotype polymorphism may be associated with the development of atherosclerotic plaques. Specifically, it has been shown that DD and ID genotypes manifest higher numbers of diseased coronary vessels, whereas genotype I shows smaller numbers of atherosclerotic lesions [95]. The D allele of the ACE gene was also found to be a strong and independent risk factor for coronary artery disease in patients with non-insulin-dependent diabetes mellitus [96].

The polymorphism of the angiotensinogen M235T genotype occurs more frequently in patients with acute myocardial infarction than in the healthy control subjects [97]. The role of the specific genotype of AT1R in susceptibility of human coronary vessels to vasoconstrictory factors was determined in patients undergoing arteriography, among whom it was found that the subjects with the CC genotype of AT1R manifest significantly greater responsiveness to methylergonovine maleate, which is a potent vasoconstrictor [98].

### 2.6. Interaction of Angiotensin Peptides with Insulin

#### 2.6.1. Interaction of Angiotensin II with Insulin

Experimental and clinical studies provide evidence for the complex interactions between angiotensin peptides and insulin in the regulation of the coronary blood flow and the myocardial function [6,99]. Thus, prolonged (7 weeks) hyperinsulinemia decreased the expression of AT1R and increased the expression of AT2R in the atrium of SD rats, whereas it enhanced the expression of both types of these receptors in the left ventricle [99]. Moreover, hyperinsulinemia elicited an increase of the LV mass and relative wall thickness and reduced the stroke volume and cardiac output. These changes were associated with the hypertrophy of myocytes, interstitial fibrosis, and increased phosphorylation of IRS-1, ERK 1/2, MEK1/2, Akt, and PI3K. It is likely that insulin could cooperate with the sympathetic system, because its effects could be significantly attenuated by the application of metoprolol, an antagonist of beta-adrenergic receptors [99].

On the other hand, Ang II was found to participate in the development of insulin resistance and endothelial dysfunction in atherosclerosis, whereas the inhibition of ACE exerted positive effects in animal models of cardiometabolic syndrome [100]. In SD rats maintained on a fructose-rich diet, which promotes the development of hypertension and insulin resistance, the administration of the ACE inhibitor or AT1R blocker (Olmesartan) significantly reduced blood pressure, improved insulin sensitivity, and reduced adipocyte size [101]. Further studies revealed that blockade of AT1R in insulin-treated rats significantly potentiated NO-mediated vasodilation [102].

#### 2.6.2. Interaction of Ang-(1-7) with Insulin

Ang-(1-7) appears to interact with insulin in the opposite way to Ang II. The subcutaneous (sc) infusion of Ang-(1-7) was found to reduce insulin resistance, hypertriglyceridemia, obesity, and hepatic fat accumulation in rats maintained on a high fructose/low-magnesium diet, which imitates the metabolic syndrome [103]. Similarly, the sc administration of Ang-(1-7) improved insulin sensitivity in C57BL/6J mice maintained on a high-fat diet and this effect was associated with increased glucose uptake into the skeletal muscle [104]. Ang-(1-7) also exerts other effects that may potently contribute to the regulation of metabolism via insulin. For instance, it has been found that it induces the proliferation of pancreatic β-cells, increases insulin secretion, ameliorates the sensitivity of skeletal muscles and adipose tissue to insulin, and improves the metabolic function of the liver [105].

#### 2.6.3. Central Effects of Insulin

It should be noted that insulin receptors and insulin-like growth factor 1 receptors are present in the central nervous system in the regions involved in the regulation of the sympathetic nervous system, metabolism, and blood pressure [106,107]. The systemic and central administration of insulin into the RVLM increases lumbar sympathetic nerve activity and this effect can be reversed by blockade of glutamatergic NMDA receptors [106]. Therefore, it is likely that the direct interaction of insulin with angiotensin peptides in the heart may be potentiated by its intra-brain action, promoting the stimulation of the sympathetic nervous system.

### 2.7. The Role of Angiotensin Peptides in the Heart in Obesity, Diabetes Mellitus, and Hypertension

Multiple experimental and clinical studies report that a high-fat diet, obesity, and diabetes mellitus significantly influence the activity of the RAS.

#### 2.7.1. The Role of Ang II in Experimental Obesity, Diabetes Mellitus, and Hypertension

Enhanced ACE activity associated with diminished responsiveness to the vasodilatory action of bradykinin was found in the coronary arterioles of obese rats maintained on a high-fat diet [108]. The treatment of type 2 diabetic KK-Ay mice (an experimental model of T2DM) with an AT1R antagonist (valsartan) significantly improved insulin sensitivity and reduced the plasma glucose level. This was associated with the phosphorylation of IRS and with the translocation of GLUT 4 to the plasma membrane. In addition, the application of valsartan reduced the expression of TNF-α and the production of superoxide. The treatment of KK-Ay mice with an AT2R antagonist (PD123319) did not have significant effects on insulin sensitivity in this study [109]; however, positive effects on metabolism by means of AT2R were found in another study on KK-AY mice, in which the intraperitoneal injections of AT2R agonist (compound 21, C21) reduced insulin resistance [110].

Experiments on db/db mice, which is another model of T2DM, provided evidence that activation AT1R plays a role in the remodeling of coronary vessels in diabetes mellitus. Thus, 16-week-old db/db mice manifested hyperglycemia, hyperlipidemia, obesity, insulin resistance, and coronary remodeling [111], whereas, after prolonged treatment with losartan in the dose which does not alter systemic blood pressure, the mice showed a significantly decreased number of VSMC, the reduced remodeling of the coronary vessels, and increased coronary flow reserve [112].

There is evidence for the enhanced involvement of angiotensin peptides in the development of insulin resistance and in the generation of oxidative stress in hypertension. In a model of Dahl-salt sensitive hypertension, the hypertensive animals manifested insulin resistance, increased the expression of AT1R mRNA and protein, and impaired Ach-induced endothelium-dependent relaxation (EDR). The administration of an AT1R blocker (candesartan) or an antioxidant compound (tempol) reduced arterial blood pressure and insulin resistance and normalized EDR and O_2_^-^ in the wall of the aorta [113]. Furthermore, SHR rats that were maintained on a high-fat diet and were developing type II diabetes (SHRDI) manifested body-weight loss, hyperglycemia, reduced plasma insulin levels, decreased myocardial and brain capillary vascularization, and enhanced oxidative stress. The harmful effects of diabetes in SHRDI rats could be prevented by the chronic inhibition of ACE with enalapril or by blockade of an AT1R antagonist (olmesartan) [114]. Obesity, increased blood pressure, myocardial hypertrophy, and interstitial fibrosis, associated with elevated levels of ERK, PI3K, and Tyr-phosphorylated β-insulin receptor subunit (βIR) and with decreased myocardial JNK expressions, were observed in Wistar-Kyoto rats fed with a hypercaloric diet for 30 weeks. Several of these disorders, such as dyslipidemia and insulin resistance, could be efficiently reduced by the administration of losartan in drinking water [115].

It is likely that some of the detrimental effects of Ang II in obesity and diabetes mellitus are mediated by aldosterone, which closely cooperates with RAS and participates in the regulation of insulin secretion. Experiments on isolated pancreatic islets revealed that aldosterone acting on mineralocorticoid receptors (MR) enhances glucose-induced insulin secretion and ROS production. The activation of MR also decreased the sensitivity to insulin in adipocytes and in skeletal muscles [116], and promoted the development of inflammation, oxidative stress, lipid disorders, and insulin resistance, thereby impairing vascular insulin metabolic signaling [15,117,118].

#### 2.7.2. The Role of Ang II in Human Obesity, Diabetes Mellitus, and Hypertension

The activity of ACE is enhanced in the coronary arterioles of obese human patients [108]. Measurements of insulin sensitivity index (ISI) and evaluations of HOMA-IR in patients with impaired glucose tolerance (IGT) and in diabetes mellitus revealed that the administration of AT1R antagonist valsartan effectively reduces resting fasting insulin levels, elevates ISI and adiponectin levels, and decreases HOMA-IR and high sensitivity C-reactive protein (hsCRP) in the IGT group [119].

There is evidence that patients with prediabetes and type 2 diabetes have elevated pericardial and peri-aortic adipose tissue [120]. It has been shown that the production of prohypertensive components of the RAS, in particular angiotensinogen mRNA and ACE mRNA, is elevated in the epicardial adipose tissue of obese patients [51]. Elevated expressions of ACE2 and *ADM17* (a disintegrin and metalloproteinase 17) genes have been found in the epicardial fat of patients with type II diabetes mellitus and in subjects with obesity [52]. As ADM17 is a membrane-bound enzyme, which participates in the proteolytic cleavage of proinflammatory cytokines, such as TNF-α [121], it is possible that *ADM17* elevation in obesity and in type II diabetes may signalize inflammatory processes and other detrimental changes, which appear in the heart. in these diseases [122].

#### 2.7.3. The Role of ACE2 and Ang-(1-7) in Obesity, Diabetes Mellitus, and Hypertension

Experimental evidence from animals maintained on high-fat diets strongly suggests that ACE2 and Ang-(1-7) may partly oppose the effects of the activation of AT1R. Studies on ACE2 mutant (ACE2KO) mice and wild type mice provided evidence that a high-fat diet causes glucose intolerance, myocardial insulin resistance, cardiac steatosis, lipotoxicity, and the development of the proinflammatory phenotype of the adipose tissue. These effects were associated with a decrease in cardiac adiponectin, which is an anti-inflammatory adipokine of adipocytes, and could be partly removed by an administration of Ang-(1-7) that lasts for 4 weeks [123]. In the obese mice maintained on a high-fat diet, the administration of Ang-(1-7) for 28 days enlarged brown adipose tissue, upregulated thermogenesis, improved impaired glucose homeostasis, and enhanced expression UCP1 uncoupling protein-1 (UCP-1) in the brown adipose tissue [124]. 

The regulation of coronary blood flow and cardiac contractility by angiotensin peptides is summarized in Table 1. 

## 3. Vasopressin System

### 3.1. Components of Vasopressin System

Arginine vasopressin is synthesized in the neuroendocrine cells of the hypothalamus, located mainly in the supraoptic nucleus (SON), the PVN, and the suprachiasmatic nucleus (SCN) [125]. In addition, immunocytochemical and functional studies provided evidence for the synthesis of AVP in the heart and lungs [126,127]. In most mammals, AVP is a chief active peptide composed of Cys-Tyr-Phe-Gln-Asn-Cys-Pro-Arg-Gly-NH2. The AVP gene is found in 20 chromosomes and consists of exons encoding the sequence of the 145-aminoacid polypeptide precursor-forming N-terminal signal peptide, and sequences of AVP, neurophysin II (NPII), and copeptin. Copeptin is a glycopeptide, composed of 39 aminoacids released in equimolar quantities as AVP and is frequently used due to its more stable biomarker molecule [128]. Measurements of copeptin have been included in ESC guidelines on the management of non-ST elevation myocardial infarction [129].

Actions of AVP are mediated by V1 receptors (V1R), which include V1a receptors (V1aR), and V1b receptors (V1bR), and by V2 receptors (V2R), and oxytocin receptors (OTR) [20,130]. V1aR are found in the brain, the spinal cord, the heart and vessels, the kidneys, the lungs, and the digestive system (mainly in the liver and the pancreas) [131,132,133]. V1bR are present in the pituitary, the brain, the pancreatic gland, and the lungs [132,134]. Thus far, V2R have been located mainly in the kidney [131,135,136]; however, there is also some evidence for their presence in the heart [137].

### 3.2. Regulation of Vasopressin Release

Under physiological conditions, the release of AVP is regulated largely by changes of blood osmolality. Among other stimulators are stress, hypovolemia, hypotension, hypoxia, hypoglycemia, and several neuroactive factors, whose significance increases in cardiovascular and metabolic diseases. Angiotensin peptides, in particular Ang II, and Ang-1-7 are potent stimulators of AVP secretion [9]. Furthermore, experiments on rats have shown that the intracerebroventricular (ICV) infusion of insulin increases plasma AVP in a dose-dependent manner and that this effect is not associated with systemic hypoglycemia [138]. In addition, hypoxia and hypoglycemia are able to stimulate AVP release, whereas vasopressin elevates blood glucose levels and increases brain glucose retention. There is evidence that these effects occur with the engagement of carotid chemoreceptors and NTS [139,140,141]. In experiments on conscious freely moving rats, it was found that the central administration of glucagon-like peptide-1 (7-36) amide (GLP1) activates the parvocellular and magnocellular neurons of the PVN and SON and elevates plasma AVP level [142]. In the PVN and SON, the vasopressinergic neurons and GLP-1 receptors are co-expressed [143].

#### Inhibitory Role of Nitric Oxide in Vasopressin Release

Nitric oxide plays significant role in the regulation of AVP secretion. NO released in the PVN inhibits the secretion of AVP and plays an essential role in buffering pressor responses that are exerted by other hypertensive compounds, such as Ang II and endothelin-1 (ET-1) [63,144]. It is worth noting that the neuronal NO synthase (nNOS) is co-expressed with AT1R in the PVN and SON neurons and that Ang II stimulates the release of AVP and upregulates AVP mRNA levels in the PVN and SON neurons; these effects are potentiated by the inhibition of nNOS activity [145]. Earlier studies suggested that during LPS-induced endotoxemia, the secretion of AVP is inhibited by NO and CO [146,147,148]. More recently, experiments on the hypothalamic explants of rats provided evidence that NO reduces AVP release, whereas CO and hydrogen sulfide (H_2_S) can exert the opposite effect [149]. 

It is also likely that NO may inhibit the local cardiac release of AVP and buffer the vasoconstrictory action of this peptide in the heart [126]. In support of this assumption, experiments on pressure-overloaded hearts of the rat in which AVP mRNA and AVP peptide were detected mainly in endothelial cells and vascular smooth muscle cells [126]. 

### 3.3. Cardiac Effects of Vasopressin

#### 3.3.1. Experimental and Human Studies

Evidence for the involvement of vasopressin in the regulation of the healthy heart comes mainly from experimental studies. Earlier research provided evidence that AVP constricts coronary vessels with diameters less than 100 μm, whereas it does not exert significant effects or can even dilate larger vessels [150,151,152]. Experiments on the isolated hearts of male Long Evans rats showed that the application of AVP significantly reduced coronary blood flow and myocardial oxygen consumption, and that these effects were markedly attenuated during acute hypoxia. The latter finding suggested that hypoxia may significantly modulate the vasoconstrictory potency of AVP in the heart [153]. It has been shown that the vasoconstrictory effect of vasopressin on coronary vessels is mediated mainly by V1R (Figure 2) and that this can be demonstrated in humans [20,154]. However, it should be noted that coronary vasoconstriction was also observed in patients with spastic angina after the application of desmopressin (DDAVP), which is a selective agonist for V2R [155].

#### 3.3.2. The Interaction of Vasopressin with NO and Endocannabinoids in the Heart

Experimental studies provide evidence that the regulation of the coronary blood flow by vasopressin is significantly modulated by locally released NO. It has been shown that AVP and its V1a agonist [Phe2Ile3proOrn8] vasopressin significantly increase iNOS mRNA and NO production, as well as cytosolic free Ca^2+^ level in cultures of neonatal rat cardiac myocytes stimulated by IL-1β [156]. Studies on the isolated coronary arteries of monkeys demonstrated that acting on V1a receptors AVP can elicit vasodilation mediated by NO [157]. Further studies performed on the hearts of SD rats showed that AVP is able to stimulate inducible NO synthase (iNOS), nuclear factor kappa-B (NF-kappa B), and collagen synthesis. The results suggested that in the heart, AVP may participate in the development of cardiac fibrosis and inflammatory processes, and these effects are attenuated by the antifibrotic action of NO [158]. Negative effects of the excessive stimulation of V1aR in the heart of mice were demonstrated by Li et al. [159], who found that the selective overexpression of V1aR in the heart causes cardiac dysfunction and hypertrophy associated with ventricular dilation and an excessive activation of the Gα_q/11_-mediated pathway [159]. Among factors which are able to decrease the vasoconstrictory potency of vasopressin are cannabinoids, which are stimulating CB1 receptors (for instance anandamide) [160].

### 3.4. Vasopressin in the Heart in Cardiovascular Diseases

Substantial evidence indicates that the responsiveness of coronary vessels to vasopressin is altered under pathological conditions.

#### 3.4.1. Experimental Studies

Measurements of CBF in conscious dogs with an experimentally induced area of hypoperfusion of the collateral-dependent myocardium showed that the systemic infusion of AVP during physical exercise decreased CBF in the collateral zone and that this effect was associated with an increase of resistance in the transcollateral and small coronary vessels. Under control conditions, the exercise did not have any significant effects on these parameters [161]. Recent studies have shown that during cardiac ischemia, AVP may play a beneficial role through the action exerted in mitochondria. For instance, experiments on male Wistar rats subjected to 30 min of ischemia followed by 120 min of reperfusion showed that the administration of AVP decreased the sensitivity of the mitochondrial permeability transition pore (MPTP) to the action of MPTP openers, thereby protecting mitochondria from swelling. In addition, the infusion of AVP significantly reduced the infarct size and plasma levels of lactate dehydrogenase, creatinine kinase-MB, and malondialdehyde [162].

#### 3.4.2. Clinical Studies

The administration of AVP in patients with catecholamine-resistant vasodilatory shock exerted positive effects including elevations of blood pressure, the stroke volume index, and the left ventricle stroke work index [163,164].

The role of vasopressin in the regulation of coronary blood flow and cardiac contractility is summarized in Table 2. 

### 3.5. Role of Vasopressin in Obesity and Diabetes Mellitus

#### 3.5.1. Clinical and Experimental Evidence for the Regulation of Glycemia by Vasopressin

Several studies provide evidence that vasopressin and insulin cooperate in the regulation of glycemia. It was found that the withdrawal of insulin in patients with diabetes mellitus elevated plasma vasopressin concentration [165], whereas the administration of AVP elicited hyperglycemia, which was associated with the stimulation of the sympathetic system [144,166]. Moreover, it has been shown that hypoglycemia stimulates the release of AVP and that the glucosensitive receptors engaged in the regulation of AVP release are located inside the blood–brain barrier [167]. The ICV infusion of insulin in doses that do not cause systemic hypoglycemia increased the secretion of AVP in a dose-dependent manner [138]. There is evidence that AVP participates in the regulation of glycemic responses during hypoglycemia. For instance, the central administration of vasopressin in rats was found to reduce the hyperglycemic and hyperglucagonemic responses to hypoglycemia induced by the administration of 2-deoxy-D-glucose [168].

#### 3.5.2. The Role of Vasopressin in the Regulation of Insulin and Glucagon Release

It has been shown that AVP actively participates in the regulation of insulin and glucagon release and closely cooperates with these hormones in the regulation of glycemia. Vasopressin has been detected in human and rat pancreatic extracts [169] and AVP receptors have been found in pancreatic islets of several species, including human beings. Moreover, it has been found that AVP stimulates the release of glucagon and insulin [132,170,171]. Interestingly, AVP was able to release insulin from beta pancreatic cells and glucagon from alpha pancreatic cells; however, the type of the cell stimulated was strongly dependent on the level of glycemia. At elevated glucose concentrations, AVP increased insulin secretion, whereas, at reduced glucose concentrations, it promoted the release of glucagon [172]. The latter finding suggested that AVP may play an essential role in the regulation of glucagon release during hypoglycemia. The pancreatic alpha cells secreting glucagon express high levels of the V1bR gene, and blockade of V1bR reduces the AVP-dependent release of glucagon. The effectiveness of the stimulation of glucagon release by AVP appears to be significantly impaired in patients with type 1 diabetes mellitus [173].

Frequently, hypoglycemia and stress stimulate the release of AVP together with CRH [167,174,175,176]. It has been also shown that the increased release of AVP during hypoglycemia significantly contributes to the stimulation of ACTH secretion [177,178,179]. It is likely that AVP plays an essential role in the regulation of ACTH secretion during hypoglycemia and stress, because AVP-deficient Brattleboro rats manifest reduced elevations of plasma ACTH and corticosterone levels during hypoglycemia and exposure to various types of stressors [180].

#### 3.5.3. The Role of Vasopressin in Diabetes Mellitus and Obesity

There is evidence that the insulin-dependent release of AVP may be markedly dysregulated in diabetes mellitus and obesity.

Experimental studies. Higher numbers of AVP mRNA expressing cells were found in the PVN of nonobese diabetic mice than in the PVN of nondiabetic mice and control C57B1/6 mice [181]. There is evidence that the responsiveness of coronary vessels to vasopressin may significantly differ in diabetic males and females. In this respect, it has been shown that isolated coronary vessels of control female rats and female rats with streptozotocin-induced diabetes are less sensitive to the vasoconstrictive action of AVP than the corresponding vessels of control and diabetic male rats. The study provided evidence that the reduced vasoconstrictory responses to AVP in females could result from an increased production of NO [182].

Engagement of the AVP-V1aR pathway in the regulation of glucose homeostasis was confirmed by experiments on receptor-deficient mice (V1aR(-/-), which showed significant disturbances in glucose metabolism, such as hyperglycemia, higher hepatic glucose production, and decreased liver glycogen content. The mutant mice responded with elevation of plasma AVP levels and developed overt obesity when they were maintained on a high-caloric intake, while the same diet was not effective in wild-type mice [183]. Furthermore, experiments performed on the V1aR knockout mice and double V1aR/V1bR knockout mice, as well as on obese Zucker rats, revealed that the inappropriate activation of V1R induces significant disturbances in glucose and lipid metabolism, which may have an indirect detrimental effect on glucose and lipid supply to the heart [184,185].

Studies on Zucker diabetic fatty (ZDF) rats, which serve as an experimental model of type 2 diabetes, demonstrated higher levels of AVP in the PVN and SON of ZDF rats than in the PVN and SON of Zucker lean control (ZLC) rats and the pre-diabetic rats. Thus, the results provided direct evidence that type 2 diabetes is associated with the increased secretion of AVP in the hypothalamic nuclei [186]. 

Clinical studies. It has been shown that the administration of insulin elicits significantly higher elevations of plasma AVP and oxytocin levels in diabetic patients than in healthy subjects [187]. Higher plasma copeptin levels and higher insulin plasma concentrations were found in obese patients and patients with metabolic syndromes [188,189,190]. On the other hand, the diabetic patients with asymptomatic hypoglycemia responded with smaller increases of blood AVP and lower elevations of heart rate in response to hypoglycemia than the healthy subjects. These findings suggested that the appropriate secretion of AVP may play an essential role in the appropriate sensation of glycemia [191]. The decreased expression of hepatic V1aR mRNA was found in rats with streptozotocin-induced diabetes mellitus [192]. 

#### 3.5.4. Genotypes of Vasopressin Receptors as Risk Factors of Coronary Diseases

Certain variations of the human *AVPR1A* gene are associated with specific abnormalities of metabolism. It has been found that the subjects carrying the *T* allele of rs 1042615 and male subjects carrying rs 1042615 allele demonstrate lower concentrations of triglycerides and higher fasting blood glucose concentrations than *CC* carriers [193]. It has also been reported that the allele of rs35810727 *AVPR1B* gene is associated with elevated body mass index, and that the human subjects carrying this variance of the *AVPR1B* gene are more prone to develop obesity and diabetes mellitus [194].

## 4. Summary and Conclusions

The review emphasizes the role of Ang II, Ang-(1-7), AVP, and insulin in the regulation of cardiac functions in health and in cardiovascular and metabolic diseases.

### 4.1. Current State of Knowledge

A survey of the literature provides strong evidence that angiotensin II, Ang-(1-7), vasopressin, and insulin have their specific receptors in the coronary vessels and cardiomyocytes, and that interacting with these receptors plays a crucial role in the regulation of the coronary blood flow and strength of the cardiac contractions (Figure 2). Prominent evidence draws attention to significant disturbances in the regulation of the cardiovascular system by Ang II, Ang-(1-7), and AVP in cardiovascular and metabolic disorders, especially when these diseases coexist. Cardiovascular diseases, in particular the myocardial infarction, causes a wide cascade of events, including hypoxia, necrosis, the increased production of inflammatory and necrotic factors, the overloading of the non-infarcted cardiomyocytes, the activation of the autonomic nervous system, and the stimulation of the hypothalamo–hypophysial axes. Pain and stress, which are frequently evoked by the myocardial infarction, severely intensify the post-infarct discomfort, and activate RAS, VS, and the autonomic nervous system. In addition, severe heart failure may cause the hypoxia of other organs, such as the brain, which plays a crucial role in the co-ordination of blood circulation, and of the kidneys, which are responsible for the maintenance of the water–electrolyte balance. In spite of the wide range of experimental and clinical studies, the mechanisms underlying defective cardiovascular regulation in diabetes mellitus are not sufficiently recognized. A growing number of studies indicate that the metabolic disorders, especially those associated with the inappropriate secretion of insulin, and insulin resistance and obesity cause significant disturbances in the secretion and action of angiotensin peptides and vasopressin. Acting on specific receptors, angiotensin peptides, vasopressin, and insulin can exert acute and prolonged synergistic or antagonistic effects and have an impact on multiple parameters and reactions that decide about the cardiac efficiency. As shown in Figure 1, among the cardiac responses to angiotensin peptides and vasopressin are cell proliferation [Ang II → AT1R, Ang II → AT2R], vasoconstriction [Ang II → AT1R, AVP → V1aR], angiogenesis [Ang II → AT2R], increased ischemia [Ang II → AT1R; AVP → V1aR], reduced ischemia [Ang-(1-7)], ROS generation [AngII → AT1R], increased insulin secretion [AngII → AT1R; Ang-(1-7) → MasR; AVP → V1aR], decreased insulin sensitivity [Ang II → AT1R], increased insulin sensitivity [(Ang-(1-7) → MasR], endothelin secretion [Ang II → AT1R], reduced triglyceridemia [(Ang-(1-7)], increased glucagon secretion [AVP → V1aR], and increased NO production [Ang II → AT1R, Ang II → AT2R, Ang-(1-7) → MasR, AVP → V1aR]. Cardiac failure, hypertension, and diabetes mellitus (especially type II diabetes mellitus) frequently occur together. As discussed in the review, angiotensin peptides and vasopressin play a significant role in the regulation of the secretion of insulin and tissue sensitivity to this hormone (Table 3). As shown in the summary, Table 1, Table 2 and Table 3, and Figure 1, the specific action of angiotensin peptides and vasopressin depends on the type of activated receptor and on the interactions with other regulatory compounds, such as NO, bradykinin, endothelin, and cytokines. Subjects suffering from heart failure and diabetes mellitus frequently manifest the altered synthesis of various components of RAS and VS. Therefore, the coupling of various types of cardiovascular diseases with metabolic disorders creates particularly challenging conditions, as it influences the sensitivity of cells to angiotensin peptides, vasopressin, and insulin, and thereby enhances the predisposition to the development of detrimental changes in the heart and vessels of diabetic patients.

### 4.2. Current Objections

It should be emphasized that the current knowledge of the interplay of angiotensin peptides and vasopressin with insulin is relatively modest and based namely on experiments performed on animals. In humans, access to satisfactory methods is still limited, and there are also ethical reservations. In addition, even in the experimental studies, there is still a large amount of information that is unknown regarding the synthesis and action of angiotensin peptides and vasopressin in diabetes mellitus and in other metabolic disorders. Nevertheless, current evidence allows us to conclude that cardiac failure and diabetes mellitus result in significant disturbances of the regulation of cardiac contractions and coronary blood flow via angiotensin peptides and vasopressin. It is also reasonable to assume that the individual patients suffering from the coexisting cardiovascular and metabolic diseases may respond differently to the complex pharmacological therapies that target RAS, VS, and insulin receptor signaling.

### 4.3. Future Directions

At present, it is reasonable to conclude that very individualized therapies, taking into account interactions of angiotensin peptides, vasopressin, and insulin, should be selected for patients suffering from co-existing cardiovascular and metabolic diseases. As the current knowledge is not sufficient enough to construct such treatments, future investigations should attempt to reach the following aims: (1) better examination of the mechanisms responsible for the interactions of angiotensin peptides and vasopressin with insulin at the receptor and the post-receptor level; (2) better exploration of the cellular mechanisms underlying the activation of angiotensin and vasopressin receptors in the cardiovascular disorders associated with obesity and diabetes mellitus; (3) the careful monitoring of the concentrations of angiotensin peptides, vasopressin, and insulin in patients suffering simultaneously from metabolic and cardiovascular diseases; and (4) the cautious planning of pharmacological therapies affecting RAS, VS, and insulin signaling pathways in patients suffering from both cardiovascular and metabolic disorders.

## Figures and Tables

**Figure 1 ijms-25-01310-f001:**
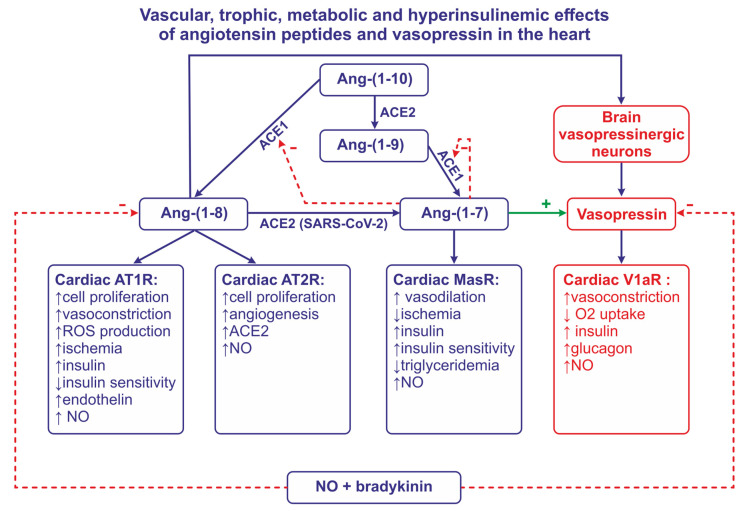
This figure illustrates the role of angiotensin converting enzyme -1 (ACE1), angiotensin converting enzyme-2 (ACE2), nitric oxide (NO), and bradykinin in formation and the release of angiotensin II (Ang-(1-8) and Ang-(1-7). This figure also summarizes effects of Ang-(1-8) and Ang-(1-7) on the release of vasopressin and insulin, and the effects of angiotensins and vasopressin on coronary vessels, insulin sensitivity, the formation of NO, and trophic and metabolic processes in the heart. Abbreviations: Ang—angiotensin, AT1R—angiotensin receptor of type 1, AT2R—angiotensin receptor of type 2, MasR—MAS receptor, SARS-CoV-2—severe acute respiratory syndrome associated coronavirus, V1aR—vasopressin receptor.

**Figure 2 ijms-25-01310-f002:**
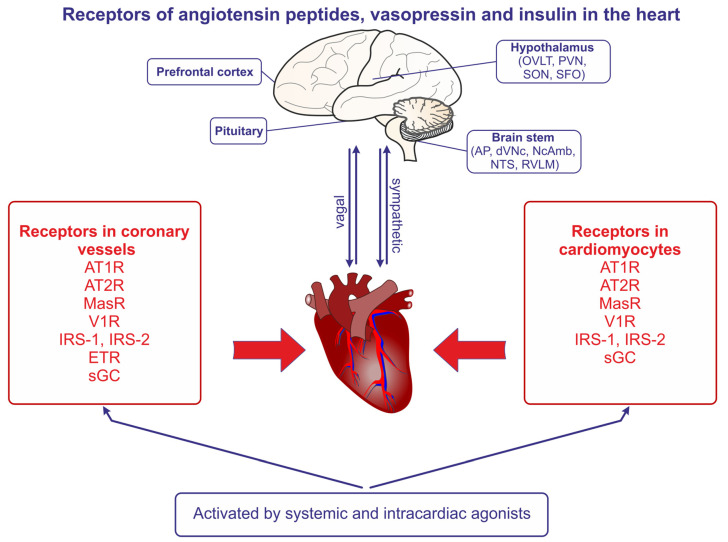
The presence of angiotensin II receptors of type 1 (AT1R) and type 2 (AT2R), the angiotensin- (1-7) receptor (MasR), the vasopressin V1 receptor (V1aR), the endothelin receptor (ETR), the nitric oxide receptor (sGC, soluble guanylyl cyclase), the insulin receptor substrate of type 1 (IRS-1), and the insulin receptor substrate of type 2 (IRS-2) in the coronary vessels and the cardiomyocytes of the heart. The regions of the brain expressing receptors of angiotensins and vasopressin are also indicated, and the bilateral sympathetic and vagal connections between the brain and the heart are shown. Abbreviations: AP—area postrema, dVNc—dorsal nucleus of vagus nerve, NcAmb—nucleus ambiguus, NTS—nucleus of the solitary tract, OVLT—organum vasculosum of the lamina terminalis, PVN—paraventricular nucleus, RVLM—rostral ventrolateral medulla, SFO—subfornical organ, SON—supraoptic nucleus.

**Table 1 ijms-25-01310-t001:** The regulation of coronary blood flow and cardiac contractility by angiotensin peptides.

Angiotensin	Subject	Receptor	Effect	References
Ang II	Rodent	AT1R	Cardiomyocytes: - Cardiac hypertrophy - Mitochondrial damage - ROS production	[7,22,24,25,26]
Ang II	Human and rodent	AT1R	Coronary vessels: Direct: - Coronary constriction - VSMC proliferation - Cell migration	[27,28,29,30,31,72,73,79,101]
			Indirect, (via *NO &Bk*): - Vasodilation - Angiogenesis	[39,40,41,42,45]
Ang II	Human and rodent	AT2R	Coronary vessels: - CES proliferation - VSMC migration - Inhibition of TNFα	[37,38,84,88]
Ang II	Rodent	AT1R	Central effects: Sympathetic stimulation - AVP secretion - Sodium appetite and thirst	[9,61,62,63,77]
Ang-(1-7)	Human & rodent	MasR	Coronary vessels: Direct: - Inhibition of ACE1 - Indirect (via *NO and Bk*) - Vasodilation	[53,54,55,56,57,58,59,103]

Abbreviations: Ang—angiotensin; ACE1—angiotensin converting enzyme of type 1, AT1R, AT2R—angiotensin II receptors; AVP—arginine vasopressin, Bk—bradykinin, CES—coronary endothelial cell, NO—nitric oxide, TNF—tumor necrosis factor, VSMC—vascular smooth muscle cell.

**Table 2 ijms-25-01310-t002:** The regulation of coronary blood flow and cardiac contractility by vasopressin.

AVP Receptor	Subject	Effect	Reference
V1aR	Rodents	Cardiac hypertrophy	[159]
		Mitochondria protection	[162]
	Primates, dogs, rodents	Coronary constriction	[20,150,151,152,154,161]
		Coronary vasodilation (*via NO*)	[156]
V2R	Human	Coronary constriction	[155]

Abbreviations: AVP—arginine vasopressin, NO—nitric oxide, V1aR, V2R—vasopressin receptors.

**Table 3 ijms-25-01310-t003:** Role of angiotensins and vasopressin in the regulation of insulin secretion and insulin sensitivity.

Peptide and Receptor	Insulin Secretion	Insulin Sensitivity	Reference
Ang II (AT1R, AT2R)	Decrease	Decrease	[100,101,109,111,113,114,115,121]
Ang-(1-7) (MasR)	Increase	Increase	[104,105,119]
AVP (V1aR, V1bR)	Increase	Decrease	[132,138,170,171,172,185,186]

Abbreviations: Ang—angiotensin, AT1R, AT2R—angiotensin receptors of type 1 and 2, AVP—arginine vasopressin, MasR—Ang-(17) receptor.

## Data Availability

Not applicable.

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
