# Peer review of "Interplay of Angiotensin Peptides, Vasopressin, and Insulin in the Heart: Experimental and Clinical Evidence of Altered Interactions in Obesity and Diabetes Mellitus"

_ijms, 2024, doi:10.3390/ijms25021310_

Round 1

Reviewer 1 Report

Comments and Suggestions for Authors

the first 3 lines of the abstract do not seem very academic and are quite simplistic, they should be rephrased because they make the first impression. 

The author states "There is evidence for intracardiac production of some of these compounds and their importance in the regulation of the cardiac tissue oxygenation" well yes, this is not revolutionary data or information, it just sounds like the authors is revolving around the same idea. 

"The local cardiac vasoactive compounds regulate CBF in concert with cardiovascular compounds inflowing from the systemic circulation" from this statement perhaps the authors should have structured the review in such a manner that there is a section about local and  another about systemic vasoactive factors.

Figure 2 shows a very rudimentary image of the heart and brain

Must the author mention a study that took place over 40 years ago ? was there not sufficient literature data about the subject that could have been mentioned instead ?

At one point the author mentions the effects of lisinopril and enalapril on the endothelium, then abruptly mentioning ace inhibitors effects on insulin, this part seems to be lacking coherence. 

Firstly from my point of view this manuscript lacks novelty in the way that it does not approach a current subject. The language is good but from my opinion it lacks coherence because it attempts to present a lot of different information which revolves around the same subject, which is mainly the angiotensin system. This is an extremely vast subject and from my opinion the author should focus on key points which should be discussed in detail rather than on many smaller points. 

Comments on the Quality of English Language

English is fine.

Author Response

Thank you very much for positive evaluation of the review and helpful remarks.

Responses

  1. The text of the Abstract has been significantly altered and the first 3 lines have been removed.
  2. Regarding lack of novelty, I hope that in each of my studies I am addressing different problems, however sometimes it is necessary to refer to previous studies, because they contain necessary information.
  3. The current version of the review has been markedly reorganized and several new subsections referring separately to the results obtained on animals and humans have been introduced. It has been indicated whether the effect was exerted by the intracardiac or systemic factor whenever it was possible to identify. I would not like to introduce additional subsections (intracardiac and systemic), because I am afraid that it would worsen readability of the text.
  4. Figures 1 and 2 have been enriched by new details.
  5. I think that it is sometimes worth to refer to old studies when they were inspiring. The questioned sentence on page 6 has been modified.
  6. To make the article more coherent, the text has been reorganized and some new subsections and summary tables have been introduced.
  7. I agree that the subject addressed in the review is extremely vast , but it was my intention to draw attention to complexity of interactions of angiotensins and vasopressin with insulin in patients suffering simultaneously from cardiovascular diseases and metabolic disorders who are treated with pharmaceutics interfering with RAS, VS, or with insulin receptors.

Reviewer 2 Report

Comments and Suggestions for Authors

A very comprehensive, detailed, and interesting review. Well-written. Figures are highly relevant, detailed, and enhance readability.

I believe that as it stands, it is worth considering, but I also think that the reading experience could be more engaging with small details that would facilitate comprehension and add dynamism to the text:

  1. 1. Just as Figures 1 and 2 focus on the renin-angiotensin-aldosterone system, similar figures should be created focusing on the vasopressin system.

  2. 2. After each of the points, a summary table should be added at the end to reinforce the acquired knowledge.

  3. 3. It would be beneficial to explore, when discussing the systems, ongoing therapies, clinical trials in planning, etc., involving these signals. While the review on basic research is comprehensive, I feel there's a lack of insight into future or ongoing clinical research contributions.

  4. 4. The conclusion section could be improved; I would add a subsection discussing possible future therapeutic targets and the benefits of each.

In any case, except for the figures and tables, I am aware that these points have been addressed in the review, although I believe they can be improved for the final version of the work. Regardless, I want to congratulate the authors on the work done.

Author Response

Thank you very much for positive evaluation of the review and helpful remarks.

Responses

  1. Information on vasopressin system are now more exposed in Figures 1 and 2 and in the Table 2.
  2. The summary tables have been introduced to the text on pages 15, 25 and 29.
  3. In the current version of review the effort has been made to better emphasize significance of findings in relevance to mechanisms of action and usefulness in pharmacological therapies. New paragraphs and subsections have been introduced to better distinguish the knowledge coming from experiments on animals and human subjects.
  4. As advised by the Reviewer the section 4. Summary and Conclusions have been divided to subsections, including the subsection 4. 3 Future directions.

Reviewer 3 Report

Comments and Suggestions for Authors

The author describes all metabolic pathways for Angiotensin Peptides, Vasopressin, and Insulin in the Heart in detail. But without any critical mind. 

First, the review reports only experimental studies for all pathways. No researcher has investigated those pathways in vivo in human hearts. So, the title should be better written. 

Second, in obesity alone, the insulin levels are increased, the blood glucose levels are normal, and dyslipidemia could exist. 

Third, in poorly controlled Diabetes Mellitus type 2, the insulin levels are decreased, there is hyperglycemia from mild to severe, and hyperlipidemia.

Fourth, in well-controlled Diabetes Mellitus type 2, the insulin levels are approximately normal, blood glucose levels are near normal, and dyslipidemia. 

So, all the conditions mentioned above are different and must be reflected correctly in experimental studies. 

The reported studies describe isolated conditions in a different metabolic milieu than those that exist in real life. The extrapolation of those experiments in human conditions cannot be made. 

Furthermore, the author describes all studies with their results only, not the characteristics of the experiment.

 Moreover, the author referred to some studies in patients without DM and obesity (e.g., Circ. J. 2010;74(11):2346-2352. doi: 10.1253/circj.cj-10-0395) and extrapolated the findings from that study. 

The author should add a critical limitation in the review with the abovementioned comments, and a title change is needed. 

The methodology is appropriate. The manuscript could be better written, and the discussion/conclusions are only acceptable if the author could change the discussion/conclusions. 

Comments on the Quality of English Language

none

Author Response

Thank you very much for positive evaluation of the review and helpful remarks.

Responses

  1. The text of the review has been reorganized and the knowledge refereeing to experiments on animals and human studies have been located in separate subsections or paragraphs.
  2. Possibility of dyslipidemia has been emphasized whenever dyslipidemia was proved by experimental evidence in studies assessing the role of angiotensins and vasopressin in diabetes mellitus or other metabolic disorder. Presence of triglyceridemia is also shown in Figure 1.
  3. Studies cited in the review were conducted mainly on animals in specific experimental environment, which significantly differ from the natural conditions. To better emphasize differences between experiments performed on animals and human subjects special subsections discussing role of angiotensins, vasopressin and insulin have been introduced to the review. The title of the study has been modified and the subsection 4.2 (Current objections have been introduced to the Section 4. Summary and Conclusions.

Round 2

Reviewer 1 Report

Comments and Suggestions for Authors

This review is not badly written, however what I am concerned with is it's lack of novelty, but it does try to make up this disadvantage by going into some level of detail and illustrating molecular mechanisms.

Comments on the Quality of English Language

English in generally fine but slight adjustments are required such as the first sentence which has been rephrased but now is not properly phrased.

Author Response

The text of the manuscript has been checked. I hope that all mistakes in spelling have been removed, and that all unclear passages have been corrected.

Reviewer 3 Report

Comments and Suggestions for Authors

no other comments

Comments on the Quality of English Language

none

Author Response

I do not see new comments and I am attaching the last corrected version of the manuscript.
